# SAM-Guided Masked Token Prediction for 3D Scene Understanding

**Zhimin Chen**[1]
Clemson University

**Liang Yang**[2]
The City University of New York

**Yingwei Li**[3]
Johns Hopkins University

**Longlong Jing**[2]
The City University of New York

**Bing Li**[✉1]
Clemson University

## Abstract

Foundation models have significantly enhanced 2D task performance, and recent works like Bridge3D have successfully applied these models to improve 3D scene understanding through knowledge distillation, marking considerable advancements. Nonetheless, challenges such as the misalignment between 2D and 3D representations and the persistent long-tail distribution in 3D datasets still restrict the effectiveness of knowledge distillation from 2D to 3D using foundation models. To tackle these issues, we introduce a novel SAM-guided tokenization method that seamlessly aligns 3D transformer structures with region-level knowledge distillation, replacing the traditional KNN-based tokenization techniques. Additionally, we implement a group-balanced re-weighting strategy to effectively address the long-tail problem in knowledge distillation. Furthermore, inspired by the recent success of masked feature prediction, our framework incorporates a two-stage masked token prediction process in which the student model predicts both the global embeddings and the token-wise local embeddings derived from the teacher models trained in the first stage. Our methodology has been validated across multiple datasets, including SUN RGB-D, ScanNet, and S3DIS, for tasks like 3D object detection and semantic segmentation. The results demonstrate significant improvements over current State-of-the-art self-supervised methods, establishing new benchmarks in this field.

## 1   Introduction

3D computer vision plays a critical role in domains such as autonomous driving and robotics. Despite its importance, this field faces significant challenges in acquiring and annotating 3D data due to the high costs and complex technical requirements involved. These challenges have led to a notable scarcity of large-scale annotated datasets. To address these issues, there has been a growing shift towards self-supervised learning (SSL) strategies, including contrastive learning and masked autoencoders (MAE), which aim to improve the learning efficiency of networks and reduce reliance on labeled data. Recently, the success of 2D foundation models like Contrastive Vision-Language Pre-training (CLIP) [41] and Segment Anything (SAM) [29] has led to significant progress in image understanding. However, large-scale 3D foundation models have not yet been proposed, primarily due to the scarcity of 3D datasets. Therefore, leveraging these powerful 2D foundation models for 3D scene understanding via self-supervised learning remains an open question.

Recent works, such as CLIP2Scene [8], Seal [32], and Bridge3D [10], have made significant progress in enhancing 3D scene understanding through the use of foundation models. CLIP2Scene effectively integrates CLIP with 3D models by implementing pixel-to-point distillation via foundation models.

38th Conference on Neural Information Processing Systems (NeurIPS 2024).

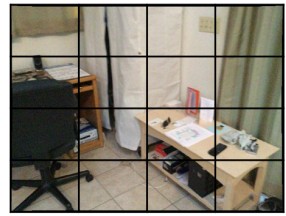 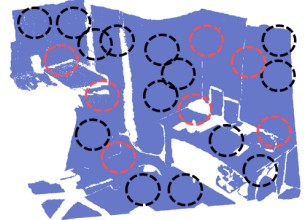 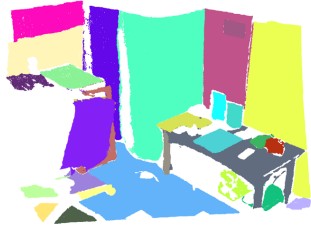

(a) Patch based 2D tokenization method.

(b) KNN-based 3D tokenization method.

(c) Proposed SAM-guided 3D tokenization method.

Figure 1: **The comparison of tokenization methods.** In Section 3.2, we present a detailed comparison of our proposed tokenization method to the previous KNN-based approach. *As shown in the red circle, the KNN-based method may inadvertently group points from different SAM regions into the same tokens, leading to potential confusion within the 3D network.* In contrast, our method effectively employs SAM masks in tokenization to ensure seamless region-level knowledge distillation, thereby avoiding these issues.

Seal distills the knowledge from 2D foundation models into the 3D network for semantic segmentation. Bridge3D introduces an innovative pre-training strategy for 3D models by utilizing features, semantic masks, and captions derived from foundation models. However, significant challenges still remain in leveraging foundation models for 3D scene understanding. Specifically, while CLIP2Scene employs point-to-text contrastive learning, it does not utilize critical region-specific information, which is essential for dense representation learning. Seal leveraged the 3D U-Net as backbone which struggles with scalability and flexibility, making it less effective for scaling and handling tasks such as detection. Bridge3D attempts to address this by using SAM-generated masks to distill vision and language representations to point tokens at the regional level. However, as illustrated in Figure 1, both Bridge3D and earlier 3D transformer-based methods employ KNN-based point tokenization strategies that can result in information conflicts during SAM-guided region-level knowledge distillation. This conflict arises when points from different SAM regions are grouped into the same 3D tokens, thereby confusing the 3D network. Furthermore, both CLIP2Scene and Bridge3D do not take into account the inherently long-tail property of 3D datasets. Giving equal weight to all samples causes the model to be predominantly driven by gradients from a few over-represented samples, leading to poor performance on under-represented samples.

To overcome these challenges, we propose a SAM-guided masked token prediction method that facilitates region-level 2D-to-3D knowledge distillation using foundation models. Unlike traditional 3D transformer methods that rely on KNN for tokenization, our approach employs masks obtained from SAM to tokenize points. This strategy effectively prevents conflicts between different regions and points, ensuring a seamless integration of point tokenization with region-level knowledge distillation. Additionally, SAM masks improve the representation of homogeneous neighboring points by more effectively leveraging boundary regularities compared to KNN-based methods. Furthermore, to address the issue of representation imbalance, we introduce a group-balanced re-weighting strategy that adjusts the distillation loss weights between 2D and 3D representations at the region level. In the self-supervised phase, where labels are absent, we utilize well-trained 2D foundation models to cluster region-level 2D representations using K-means, assigning pseudo-labels based on their cluster index. During training, we enhance the weights for tail groups while reducing them for head groups. Inspired by the recent success of masked feature prediction [56] in cross-modality learning, we introduce a two-stage masked token prediction framework. In the first stage, we perform dense region-level knowledge distillation using the SAM-guided tokenization method to transfer well-learned knowledge from the foundation model to the 3D network. In the second stage, we have the student model predict both the instance-level global embedding and the token-wise local embeddings obtained from the teacher model in the first stage based on visible 3D input patches. This approach ensures that the student model learns well-aligned and contextualized local and global representations, thereby improving its performance on downstream tasks.

We validated our methodology across multiple datasets and tasks, including SUN RGB-D [46] and ScanNet [14] for 3D object detection, and S3DIS [4] and ScanNet [14] for 3D semantic segmentation.

Our approach outperforms current state-of-the-art self-supervised learning methods, underlining the effectiveness of our proposed framework.

The key contributions of our work are summarized as follows:

- We introduce a novel two-stage SAM-guided masked token prediction framework that leverages foundation models for 3D scene understanding.

- We present a group-balanced re-weighting method for long-tail representation distillation and a SAM-guided tokenization method to seamlessly align 2D and 3D region-level features.

- Extensive experiments have been conducted to demonstrate the significance of our approach in various 3D downstream tasks.

## 2   Related Work

**3D Self-supervised Pre-training.**   The field of self-supervised learning for point clouds has witnessed substantial advancements, with researchers exploring a variety of pre-training strategies to enhance the transferability and initialization quality of networks for downstream tasks [1, 21, 11, 57, 28, 19]. These strategies range from learning the relative positions of points [43] to deploying multiple pretext tasks [22] and employing contrastive learning approaches [17, 27, 42, 49, 2, 28, 20, 18]. Innovatively, Info3D [42] applies InfoMax principles and contrastive learning to 3D shapes, enhancing feature extraction from complex geometries. PointContrast [49] performs point-level contrastive learning across transformed views of a single point cloud, promoting robustness to spatial alterations. Meanwhile, the work by Zhang [55] contrasts instance-level representations derived from different architectural processes within identical scenarios. Additionally, CrossPoint [2] pioneers a multi-modal contrastive framework that establishes 3D-2D correspondences, capitalizing on the complementary attributes of point clouds and images.

**Masked Autoencoder**   To enhance masked image modeling, the Masked Autoencoder [23] (MAE) was initially introduced for 2D images, utilizing an asymmetric encoder-decoder transformer architecture [16, 6]. This process starts with an encoder that receives a randomly masked image to extract high-level latent representations. A lightweight decoder then processes these representations, reconstructing the raw RGB pixels of the masked patches. Demonstrating exceptional performance across various downstream tasks, MAE has inspired a range of innovative adaptations. Expanding to 3D data, some adaptation methods [37, 12, 53] have been proposed to apply MAE-style pre-training to 3D point clouds. These methods involve sampling visible point tokens for the encoder and reconstructing masked 3D coordinates with the decoder. However, these 3D MAE applications have predominantly focused on masked point reconstruction. Recent studies [5, 56] have shown that masked feature prediction can be a more effective strategy for representation learning. Building on this insight, our work introduces a two-stage framework specifically designed to enable masked token prediction in 3D scene understanding, aiming to enhance the learning efficiency and applicability of MAE in complex 3D environments. This novel approach promises to push the boundaries of how deep learning models perceive and interpret three-dimensional data.

**Multi-modality Learning for 3D Scene Understanding**   Numerous studies have explored knowledge transfer from pre-trained 2D foundation models to 3D representations at the object level [24, 25, 54, 52, 38]. For comprehensive scene understanding, SlidR [44] initially employs the super-pixel technique to define regions and subsequently utilizes the InfoNCE loss for region-level contrastive learning between point cloud and 2D representations. The CLIP2Scene approach [8] leverages the MaskClip model [59] to generate dense semantic predictions. However, it lacks the capability to produce instance-specific semantic results, which is crucial for distilling object-level visual representations into 3D models. Bridge3D [10] proposes an innovative pre-training strategy for 3D models using features, semantic masks, and captions derived from foundation models. It integrates region-level knowledge distillation with a masked autoencoder for 3D scene understanding, achieving state-of-the-art performance. Despite these advancements, the KNN-based tokenization method employed in Bridge3D and other existing 3D transformer-based methods faces challenges. Specifically, the mismatch between KNN grouping and predefined mask regions can lead to representational conflicts, ultimately degrading performance. To address these limitations, we innovatively propose a SAM-guided tokenization method that seamlessly integrates region-level distillation with a

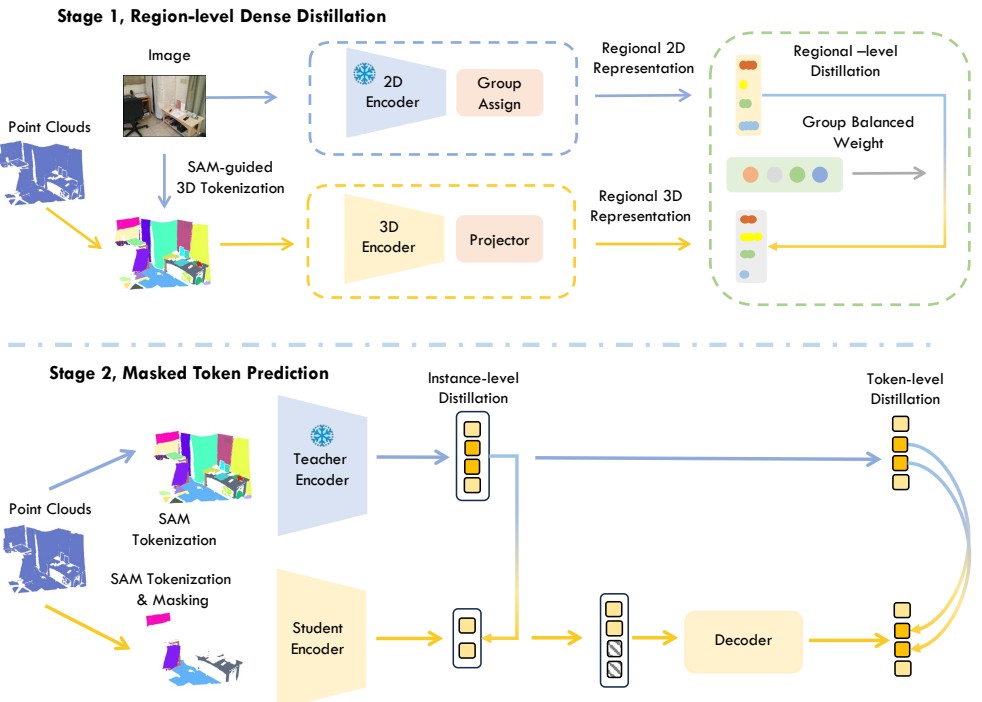

Figure 2: **Overall framework of the proposed method.** Our method introduces a two-stage masked token prediction framework for learning from foundation models. In the first stage, we input complete point clouds and leverage SAM masks to guide the point cloud tokenization, thereby seamlessly aligning the 2D and 3D region-level features for dense prediction. A group-balanced weight is applied during distillation to prevent bias towards the head representations. In the second stage, we freeze the models trained in the first stage and have the student models predict instance-level features and masked tokens obtained from the teacher models.

plain transformer architecture. This method ensures coherent region-level knowledge transfer and enhances the overall efficacy of the learning process in 3D scene understanding.

## 3  Methodology

In this section, we outline our methodology, starting with how we utilize foundation model SAM [29] to obtain masks offline. We then describe our approach to tokenize point clouds using SAM instead of KNN to address the misalignment between 2D and 3D representations. Next, we introduce the group-balanced re-weighting strategy to address the long-tail representation issue in knowledge distillation. Finally, we detail our two-stage masked token framework, which facilitates the model in learning well-aligned and contextualized representations through a process of two-stage masked token prediction.

### 3.1  Mask Generation

To enable region-level distillation, we utilized the foundation model SAM [29] to generate masks within visual images. SAM-generated masks provide comprehensive coverage of both the object and its surrounding context. This integration enables us to obtain a cohesive set of segmentation masks $\mathcal{O}_1, \ldots, \mathcal{O}_N$. To establish a precise correspondence between mask-level visual features and point tokens $\{x_i, p_i\}$, we align the point cloud tokens with the respective SAM masks, where $x_i$ and $p_i$ represent paired image and point features, respectively. This process is conducted offline, and the resulting labels are stored locally for easy access during the self-supervised training phase.

## 3.2 SAM-guided Point Tokenization

Recent state-of-the-art method Bridge3D [10] adapts the 3D plain transformer architecture for knowledge distillation from foundation models. Like other 3D transformer-based methods, Bridge3D directly tokenizes point clouds using farthest point sampling (FPS) and K-nearest neighbors (KNN) algorithms. Specifically, given a point cloud $X^i \in \mathbb{R}^{N \times 3}$ with $N$ points, FPS is used to select $n$ centroid points (CT) for forming point patches. Subsequently, KNN determines the $k$ nearest points to each centroid, forming the corresponding token $P$.

To enable efficient distillation from 2D to 3D using foundation models, Bridge3D proposes a region-level distillation strategy. However, we find that this strategy cannot effectively align 2D and 3D information when using KNN-based point tokenization. As depicted in Figure 1, KNN-based point tokenization strategies can group points from different SAM regions into the same 3D tokens. This misalignment causes information conflict, which confuses the 3D network and degrades the distillation performance.

To address this challenge, we introduce a novel SAM-guided point patch generation method tailored for multi-modality region-level knowledge distillation. We start by projecting the 3D point cloud onto corresponding 2D images. Then, we assign points to tokens based on their positions within the SAM-defined regions in the 2D images. Each patch's centroid is calculated as the average position of all points within that patch. Features are then extracted using PointNet, ensuring that each token's representation is both cohesive and regionally consistent. This methodology not only enhances the alignment between 3D points and their corresponding 2D regions but also significantly improves the performance of our knowledge distillation framework by leveraging boundary regularities provided by SAM.

## 3.3 Dense Feature Distillation

To enhance the distillation of rich representations from 2D to 3D, we build on the methodologies proposed in Bridge3D [10], utilizing region proposals from the SAM vision foundation model. Our method primarily differs from Bridge3D in our approach to aligning 2D and 3D representations. Unlike Bridge3D, which employs the traditional KNN method to generate point tokens, often resulting in the aggregation of points from disparate regions as illustrated in Figure 1, our method uses the SAM model to guide point cloud tokenization. This ensures a one-to-one correspondence between point tokens and region-level 2D features, avoiding the representation conflicts that impair 3D understanding in the Bridge3D approach. This targeted tokenization results in a comprehensive set of segmentation masks $\mathcal{O}_1, \ldots, \mathcal{O}_N$, each enriched with its corresponding textual narrative, thereby providing a deeper contextual dataset.

For detailed feature extraction, we define $E_{3D}^{\theta}$ as the trainable 3D network and $E_{2D}^{\theta}$ as the frozen pre-trained 2D encoder. The 3D network processes the point cloud, while the 2D encoder manages the corresponding images. These components generate 3D point token features $H \in \mathbb{R}^{M \times L}$ and 2D image pixel features $I_j \in \mathbb{R}^{h \times w \times L}$, where $M$ represents the number of regions, equivalent to the segmentation masks produced by SAM, and $L$ denotes the feature dimension; $h$ and $w$ are the height and width of the image features, respectively. We project the point token features to 2D space via a projection layer to obtain projected 3D features $F_{3D}$. We then pool pixel representations within the same SAM-generated region to derive region-level 2D representations $F_{2D} \in \mathbb{R}^{M \times L}$. Simultaneously, we establish correspondences between each 3D token and its matching 2D regional representation $(H_i, F_i)_{i=1}^{M}$, where $H_i$ and $F_i$ are the paired 3D and 2D regional features, ensuring a direct and meaningful alignment between the modalities. This setup allows for robust region-level feature-dense distillation, effectively training our model to better understand and interpret complex 3D scenes. The distillation process is formulated as follows:

$$F_{2D,i} = \frac{1}{\mathcal{O}_i} \sum_{i \in \mathcal{O}_j} (I_j) \tag{1}$$

$$F_{3D,i} = Proj(H_j) \tag{2}$$

Where $H_i$ represents point tokens, $Proj$ is the projection layer. The objective function is as follows:

$$\mathcal{L}_{distill} = \frac{1}{M} \sum_i^M L_1(F_{2D,i}, F_{3D,i}) \tag{3}$$

The $L_1$ is the smooth $L_1$ loss.

### 3.4 Group Balanced Re-weighting

Training models on 3D datasets, which are inherently highly imbalanced, presents significant challenges. Directly applying approaches that are not specifically designed for imbalanced datasets often results in sub-optimal network performance and thus fails to deliver satisfactory outcomes. To address this issue, recent studies, as noted by Cui et al. [13], Yu et al. [50], and Alshammari et al. [3], have introduced class-balanced loss strategies. These strategies involve recalibrating the weights in the loss function to focus more on underrepresented (tail) classes and reduce the emphasis on overrepresented (head) classes. Such adjustments aim to establish a more equitable training environment, enhancing fairness and boosting the robustness of the model.

In our 2D to 3D pre-training tasks, the lack of explicit labels complicates the identification of head and tail representations within the data. To tackle this challenge of imbalance in the representation learning stage, we propose a prototype-level re-weighting method. Leveraging the discriminative features provided by foundation models, we can directly cluster these features and use their cluster indices as pseudo-labels. Specifically, we utilize foundation models such as DINOv2 [36] or CLIP [41] to extract visual features, which are then resized to the original image dimensions using interpolation. Next, we apply max pooling across features within regions defined by SAM-generated masks to obtain region-level features. These features are grouped into $K$ clusters using KNN, categorizing them into distinct groups. Each region-level feature is assigned a group index, which we treat as a pseudo-label for applying a class-level reweighted loss. We then count the number of regions in group $i$ as $n_i$. This innovative approach allows us to effectively address the long-tail problem in representation learning by balancing the influence of each group during the learning process. $k_i = \frac{n_i - n_{\min}}{n_{\max}}$, Where $\tau_i = 1.0 - k_i$ and $w_i = \frac{\tau_i}{\sum_{j=0}^K \tau_i}$. Hence, the dense distillation loss for the first stage is:

$$\mathcal{L}_{distill} = \frac{1}{M} \sum_i^M w_i L_1(F_{2D,i}, F_{3D,i}) \tag{4}$$

### 3.5 Maksed Token Prediction

As illustrated by VideoPrism [56], latent space reconstruction is an effective method for cross-modality knowledge distillation. Inspired by this, we propose a two-stage framework to integrate latent space prediction within the MAE framework for 2D to 3D knowledge distillation. This approach differs from previous 3D masked autoencoder methods like Point-MAE [37] and Bridge3D [10], which reconstruct raw, masked inputs as their targets.

As depicted in Fig. 2, our approach involves a two-stage process. In the first stage, we perform dense feature knowledge distillation from 2D to 3D using foundation models with the proposed SAM-guided tokenization method. In the second stage, we implement a teacher-student framework. The model from the first stage serves as the teacher and is frozen for the second stage. During training, all tokens are processed by the teacher model to generate token features, and the student model is tasked with reconstructing these detailed 3D token representations using only the visible parts of the data. We introduce an instance-level distillation loss to guide the student model's learning, pushing the limits of self-supervised learning in comprehending 3D spaces.

Specifically, we send complete point tokens to the teacher models and masked point tokens to the student models. For the instance-level knowledge prediction, we pool all point token features after the teacher encoder as $F_{ins}^{teacher}$ and after the student encoder as $F_{ins}^{student}$. The student model then predicts $F_{ins}^{teacher}$ using MLP layers. The instance prediction is formulated as follows:

$$\mathcal{L}_{ins} = MSE(MLP(F_{ins}^{student}), F_{ins}^{teacher})) \tag{5}$$

| Methods | Pre-trained | SUN RGB-D | | ScanNetV2 | |
| --- | --- | --- | --- | --- | --- |
| | | $AP_{25}$ | $AP_{50}$ | $AP_{25}$ | $AP_{50}$ |
| VoteNet [39] | *None* | 57.7 | 32.9 | 58.6 | 33.5 |
| PointContrast [49] | ✓ | 57.5 | 34.5 | 59.2 | 38.0 |
| Hou et al. [26] | ✓ | - | 36.4 | - | 39.3 |
| 4DContrast [9] | ✓ | - | 38.2 | - | 40.0 |
| DepthContrast [55] | ✓ | 61.6 | 35.5 | 64.0 | 42.9 |
| DPCo [30] | ✓ | 60.2 | 35.5 | 64.2 | 41.5 |
| 3DETR [35] | *None* | 58.0 | 30.3 | 62.1 | 37.9 |
| +Plain Transformer | *None* | 57.6 | 31.9 | 61.1 | 38.6 |
| +Point-BERT[51] | - | - | - | 61.0 | 38.3 |
| +Point-MAE [37] | ✓ | - | - | 63.4 | 40.6 |
| +MaskPoint [31] | ✓ | - | - | 63.4 | 40.6 |
| +ACT [15] | ✓ | - | - | 63.5 | 41.0 |
| +PiMAE [7] | ✓ | 59.9 | 33.7 | 63.0 | 40.2 |
| +Bridge3D [10] | ✓ | 61.8 | 37.1 | 65.3 | 44.2 |
| +Ours | ✓ | **63.5(+1.7)** | **39.5(+2.4)** | **68.2 (+2.9)** | **48.4(+4.2)** |
| GroupFree3D [33] | *None* | 63.0 | 45.2 | 67.3 | 48.9 |
| +Plain Transformer | *None* | 62.2 | 45.0 | 66.1 | 48.3 |
| +Point-MAE [37] | ✓ | 63.9 | 46.1 | 67.4 | 49.8 |
| +PiMAE [7] | ✓ | 65.0 | 46.8 | 67.9 | 50.5 |
| +Bridge3D [10] | ✓ | 67.9 | 48.5 | 69.1 | 51.9 |
| +Ours | ✓ | **68.9(+1.0)** | **52.1(+3.6)** | **72.3(+3.2)** | **55.7(+3.8)** |

Table 1: **3D object detection results on ScanNet and SUN RGB-D dataset.** We adopt the average precision with 3D IoU thresholds of 0.25 ($AP_{25}$) and 0.5 ($AP_{50}$) for the evaluation metrics.

We use the global features of the student model with only visible inputs to predict the global features of the teacher model with complete inputs. Additionally, we employ a token-level prediction loss to ensure that the student models can predict the masked tokens obtained from the teacher model's decoder.

$$\mathcal{L}_{token} = \frac{1}{N_m} \sum_{i=1}^{N_m} MSE(F_i^{student}, F_i^{teacher}). \tag{6}$$

Where $N_m$ is the number of masked tokens. The effectiveness of this learning setup is evaluated through a defined reconstruction loss, ensuring high precision in the alignment between the student's and teacher's outputs. Notably, we continue to employ the SAM-guided tokenization method to facilitate this process. The final loss for the second stage is formulated as:

$$\mathcal{L}_{final} = \mathcal{L}_{ins} + \mathcal{L}_{token} \tag{7}$$

## 4 Experiments

This section begins with an overview of the pre-training and fine-tuning configurations for our method. Subsequently, we demonstrate the method's effectiveness through its application to several prominent downstream tasks, such as 3D object detection and 3D semantic segmentation. Finally, we present comprehensive ablation studies to validate the impact and contribution of each component within our approach.

### 4.1 Self-supervised Pre-training and Fine-tuning

**Pre-training.** In our pre-training stage, we leverage the ScanNet dataset [14], aligning with approaches adopted in prior research [39, 55] to obtain the corresponding image and point cloud pairs. ScanNet is an indoor dataset that includes approximately 1,500 scans derived from 2.5 million RGB-D frames. We follow the official protocol for training/validation splits, extracting 78,000 frames from the training subset by sampling one frame every 25 frames to construct our dataset. For the

| Methods | Pre-trained | S3DIS | | ScanNetV2 | |
| | | $mIoU$ | $mAcc$ | $mIoU$ | $mAcc$ |
| --- | --- | --- | --- | --- | --- |
| SR-UNet [49] | *None* | 68.2 | 75.5 | 72.1 | 80.7 |
| PointContrast [49] | ✓ | 70.9 | 77.0 | 74.1 | 81.6 |
| DepthContrast [55] | ✓ | 70.6 | - | 73.1 | - |
| Hou et al. [26] | ✓ | 72.2 | - | 73.8 | - |
| Standard Transformer [51] | *None* | 60.0 | 68.6 | - | - |
| PointBert [51] | ✓ | 60.8 | 69.9 | - | - |
| PViT [40] | *None* | 64.4 | 69.9 | - | - |
| PViT+Pix4Point [40] | ✓ | 69.6 | 75.2 | - | - |
| Plain Transformer | *None* | 61.1 | 67.2 | 67.3 | 73.1 |
| +Point-MAE [37] | ✓ | 64.8 | 70.2 | - | - |
| +Bridge3D [10] | ✓ | 70.2 | 76.1 | 73.9 | 80.2 |
| +Ours | ✓ | **71.8 (+1.6)** | **78.2(+2.1)** | **75.4(+1.5)** | **81.5(+1.3)** |

Table 2: **3D semantic segmentation results on S3DIS and ScanNet dataset.** We adopt the mean accuracy (mAcc) and mean IoU (mIoU) for the evaluation metrics.

optimization process, we utilize the AdamW optimizer [34] throughout both stages of our training, starting with a base learning rate of 0.001 and a weight decay set at 0.05. Our data is processed in batches of 64. During the second stage of training, we increase the masking ratio ($r_w$) to 60%. To further enhance the training dynamics, we implement a cosine learning rate scheduler coupled with a drop path rate of 0.1 and include a warm-up phase of 10 epochs to facilitate a smooth adjustment to the training conditions. For the 3D backbone encoder, we adopt the plain transformer structure used in Bridge3D [10]. On the image processing side, we employ the DINOV2 ViT-B model [36] to extract features. To adapt these features back to the original input size, we apply interpolation-based up-sampling techniques. The training is conducted using four A100 GPUs.

**Fine-tuning.** Following Bridge3D [10], we remove the decoders used in pre-training and introduce task-specific decoders for various downstream tasks. A key distinction between our fine-tuning approach and Bridge3D is the use of SAM-guided tokenization to generate tokens, rather than the traditional KNN-based tokenization methods employed by Bridge3D. Additionally, for detection tasks, we do not introduce new query embeddings. Instead, we use the tokens generated through SAM-guided tokenization as queries for self-attention. These tokens represent features of homogeneous neighboring regions defined by precise boundary regularities from SAM, making them suitable for 3D object detection tasks. Apart from these adjustments, we adhere to the same fine-tuning settings as Bridge3D for downstream tasks.

## 4.2 Results on Downstream Tasks

**Object Detection.** We demonstrate the generality of our proposed method by conducting pre-training on the indoor ScanNetV2 dataset [14] and subsequently fine-tuning it for object detection tasks in both the ScanNetV2 and SUN-RGBD [58] datasets. Building upon the baseline detection methods 3DETR [35] and GroupFree3D [33], our method significantly outperforms the previous state-of-the-art method Bridge3D. Specifically, our performance surpasses Bridge3D by 2.9 and 4.2 in $AP_{25}$ and $AP_{50}$ using the 3DETR baseline, and by 3.2 and 3.8 in $AP_{25}$ and $AP_{50}$ using the GroupFree3D baseline on the ScanNetV2 dataset. This consistent improvement over Bridge3D underscores the efficacy of our method in learning advanced 3D representations for object detection, indicating its potential for enhancing 3D scene understanding tasks.

**Semantic Segmentation.** In Table 2, we present the semantic segmentation results on the S3DIS [4] and ScanNet [14] datasets. Although Bridge3D [10] has improved the plain transformer baseline by a large margin, our method still outperforms Bridge3D by 1.6 and 1.5 $mIoU$ on the ScanNet and S3DIS datasets, respectively. It should be noted that Bridge3D utilizes foundation models with both 2D and text modalities, incorporating complex architectures. In contrast, our method, which utilizes only 2D foundation models, still outperforms Bridge3D in 3D semantic segmentation tasks. This demonstrates the efficiency of our proposed knowledge distillation strategy for enhancing 3D representation learning for semantic segmentation.

| Dense Distillation | Masked Token Prediction | Balanced Re-weight | SAM-Guided Tokenzie | ScanNetV2 $AP_{25}$ | $AP_{50}$ | S3DIS $mIoU$ | $mAcc$ |
|:---:|:---:|:---:|:---:|:---:|:---:|:---:|:---:|
|  |  |  |  | 61.1 | 38.6 | 61.1 | 67.2 |
| ✓ |  |  |  | 62.4 | 41.7 | 66.2 | 71.3 |
| ✓ | ✓ |  |  | 64.5 | 44.3 | 68.7 | 74.1 |
| ✓ | ✓ | ✓ |  | 66.0 | 46.1 | 69.7 | 75.9 |
| ✓ | ✓ |  | ✓ | 67.1 | 47.0 | 70.9 | 77.0 |
| ✓ | ✓ | ✓ | ✓ | **68.2** | **48.4** | **71.8** | **78.2** |

Table 3: **The effectiveness of each component.** Ablation study on the effectiveness of each component on 3D object detection and semantic segmentation tasks.

|  | ScanNetV2 $AP_{25}$ | $AP_{50}$ | S3DIS $mIoU$ | $mAcc$ |
|:---|:---:|:---:|:---:|:---:|
| Stage 1 | 65.2 | 45.1 | 69.1 | 75.3 |
| Stage 1 + MTP in same stage | 66.0 | 46.3 | 69.9 | 76.1 |
| Stage 1 + Stage 2 (Ours) | **68.2** | **48.4** | **71.8** | **78.2** |

Table 4: **The effectiveness of Stage.** Ablation study on the effectiveness of a two-stage framework on 3D object detection and semantic segmentation tasks. MTP here represents the masked token prediction

## 4.3 Ablation Study

**The Effectiveness of Each Component.** As illustrated in Table 3, the results effectively demonstrate the advantages of each component incorporated into our comprehensive framework. The detailed ablation study reveals that incorporating dense distillation significantly enhances 3D representation learning and improves overall system performance. Additionally, the implementation of a two-stage masked token prediction enables student models to learn well-aligned and highly contextualized representations across different modalities, thereby further enhancing overall system performance. Moreover, the introduction of balanced re-weighting mechanisms significantly boosts network performance by effectively mitigating the long-tail distribution challenge, which is inherently problematic in 3D datasets. Finally, the integration of SAM-guided tokenization marks the most substantial improvement within our framework, as it seamlessly aligns 2D and 3D features, thus avoiding potential conflicts and discrepancies in information transfer. In conclusion, each component of our proposed method is designed to be complementary to the others; their combined application not only achieves optimal results but also markedly enhances performance across both 3D object detection and semantic segmentation tasks, demonstrating the robustness and efficacy of our approach.

**The Effectiveness of Each Stage.** Table 4 underscores the effectiveness of our proposed two-stage method, which initiates with the distillation of dense representations from 2D foundation models into 3D models during the first stage. This initial phase of our study is meticulously designed to assess whether it is imperative to employ a teacher model or if the strategy of predicting masked 2D features directly within the first stage could potentially achieve superior or equivalent performance. The results derived from this meticulous ablation study suggest that merely combining the initial stage with masked token prediction yields only modest improvements in overall performance. However, a significant enhancement is observed with the addition of the second stage, which more thoroughly integrates our structured teacher-student framework into the process. This marked improvement distinctly highlights the critical importance of the teacher-student design within our innovative approach, confirming that the detailed and layered integration of these essential elements is absolutely vital for obtaining well-learned, robust representations that are crucial for effective 3D scene understanding tasks.

## 4.4 Apply on SOTA 3D Detectors

We recognize that applying our method to state-of-the-art detection models can further demonstrate its generality and robustness. Therefore, we applied our approach to two leading 3D detection methods: CAGroup3D [47] and VDETR [45]. Due to the specifically designed BiResNet backbone

| Methods | Pre-trained | ScanNet ($AP_{25}$) | ScanNet ($AP_{50}$) |
|---|---|---|---|
| CAGroup3D [4] | None | 75.1 | 61.3 |
| + Ours | ✓ | **76.5** | **62.4** |
| VDETR [5] | None | 73.6 | 60.1 |
| + Ours | ✓ | **75.8** | **63.0** |

Table 5: 3D object detection results on ScanNet dataset based on CAGroup3D and VDETR.

| Methods | ScanNet ($AP_{25}$) | ScanNet ($AP_{50}$) | ScanNet ($mIoU$) |
|---|---|---|---|
| Scratch | 61.1 | 38.6 | 67.3 |
| CLIP2Scene [8] | 62.0 | 40.1 | 69.2 |
| Seal [32] | 62.7 | 41.3 | 70.3 |
| PPT [48] | 62.8 | 42.1 | 70.9 |
| **Ours** | **68.2** | **48.4** | **75.4** |

Table 6: Comparison with other pre-training methods with different backbones on ScanNet dataset in 3D detection and semantic segmentation tasks.

used by CAGroup3D, we were able to apply only our SAM-guided knowledge distillation and representation re-weighting techniques to it. For VDETR, which reports results using both a modified ResNet34 encoder and a plain transformer encoder, we replaced its encoder with our pre-trained encoder and compared the performance to the transformer backbone results reported in the original paper. The experimental results presented in Table. 5 show that our pre-training strategy enhances the performance of these state-of-the-art 3D detection models. Moreover, the performance improvement in VDETR, facilitated by our proposed SAM-guided tokenization and two-stage masked autoencoder, is greater than that observed in CAGroup3D, highlighting the effectiveness of our approach.

### 4.5 Results Comparison with Pre-Training Methods for Other Backbones

In the main paper, we did not compare our results with Seal [32], PPT [48], and CLIP2Scene [8] as they use 3D-UNet as the backbone and are exclusively fine-tuned for 3D semantic segmentation tasks. Most previous methods operate at the object-level or scene-level using transformer-based 3D models. To demonstrate the effectiveness of our approach specifically designed for transformers, we adapted the methodologies of Seal, PPT, and CLIP2Scene to the transformer structure, applying the same experimental settings as our method.

As shown in Table. 6, our method achieves the best performance, highlighting the advantages of our proposed strategies. In the revised version, we will cite these papers and include a discussion of their methodologies and results. We acknowledge that including comparisons with methods using different backbones could better illustrate the effectiveness of our approach, and therefore, we have undertaken this additional evaluation. The results clearly demonstrate that our approach outperforms these existing methods, emphasizing the robustness and generalizability of our pre-training strategy.

## 5 Conclusion

In conclusion, our study addresses the challenges of aligning 2D and 3D representations and enhances knowledge distillation in self-supervised learning for 3D scene understanding. We introduced a novel SAM-guided tokenization method that aligns 3D transformer structures with region-level insights, improving distillation effectiveness. Additionally, we propose a group-balanced re-weighting strategy to address the long-tail problem. Furthermore, we introduce a two-stage masked token prediction framework, enabling the student model to predict both global instance-level embeddings and local token-wise embeddings learned from the teacher model based on visible 3D input tokens. Experiments conducted on datasets such as SUN RGB-D, ScanNet, and S3DIS demonstrate the state-of-the-art performance of the proposed method in 3D object detection and semantic segmentation tasks. Our work is expected to have no negative societal implications.

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
