# OpenReview forum: "SAM-Guided Masked Token Prediction for 3D Scene Understanding"
_NeurIPS.cc/2024/Conference — NeurIPS 2024 poster_

### Official Review · Reviewer_icGN · 2024-07-01

**Soundness:** 3
**Presentation:** 3
**Contribution:** 2
**Rating:** 5
**Confidence:** 5

**Summary:**

This paper proposes a two-stage SAM-guided pre-training method for 3D scene understanding. The authors present a group-balanced re-weighting method for long-tail representation distillation and a SAM-guided tokenization method to seamlessly align 2D and 3D region-level features. Extensive experiments on various downstream tasks show the effectiveness of the proposed pre-training method.

**Strengths:**

1. Adopt SAM to guide pre-training makes sense, since the semantic information can help model to learn high-quality patterns.
2. The results are good, surpassing previous state-of-the-art methods significantly.

**Weaknesses:**

1. For 3D object detection results, even with the proposed pre-training method, the model's overall mAP is lower than SOTA 3D detector like CAGroup3D and VDETR. Can this work be applied on these more advanced architectures? If so, supporting experimental results are required.
2. Using SAM to guide 3D pretrained has already been explored in SEAL [1], more comparison or discussion is required.
3. This paper should not only compare with transformer-based pretraining methods, but also other state-of-the-art pretrianing techniques like PPT [2].



[1] Segment Any Point Cloud Sequences by Distilling Vision Foundation Models, NeurIPS 2023
[2] Towards Large-scale 3D Representation Learning with Multi-dataset Point Prompt Training, CVPR 2024

**Questions:**

It seems the models are pretrained on independent RGB-D frames, and then finetuned on reconstructed room-level point clouds. There may be some domain gap between the single-view point clouds and the complete room-level ones. I wonder is this work applicable to the online 3D perception setting [1, 2], which is more relevant to RGB-D perception and is a more valuable setting.

[1] Fusion-aware point convolution for online semantic 3d scene segmentation, CVPR 2020
[2] Memory-based Adapters for Online 3D Scene Perception, CVPR 2024

**Limitations:**

Limitations is discussed.

---

> ### Author Rebuttal · Authors · 2024-08-06
>
> We sincerely appreciate your thorough review and valuable suggestions. We have addressed your questions as follows.
>
>
> **Q1, apply on SOTA 3D detectors.**
>
> In the main paper, we did not apply our method to other state-of-the-art detection methods because those works often incorporate specialized design structures different from the pre-training backbones, making comparisons less fair. *For pre-training tasks, comparisons are typically made with previous pre-training methods using the same structures. For example, Seal [1] only conducts training on the 3D U-Net and compares its results with previous outdoor pre-training methods.* The overall performance of Seal, as reported in the paper, is also lower than state-of-the-art 3D semantic segmentation methods like 2DPASS [2] and LidarMultiNet [3] with full supervision on the nuScenes dataset.
>
> However, we understand that applying our method to state-of-the-art detection methods can further demonstrate its generality. Therefore, we applied our approach to the state-of-the-art methods CAGroup3D [4] and  VDETR [5]. Since CAGroup3D utilizes a specifically designed BiResNet backbone, we could only apply our SAM-guided knowledge distillation and representation re-weighting method to it. For VDETR, which reports results with both a modified ResNet34 encoder and a plain transformer encoder, *we replaced its encoder with our pre-trained encoder and compared it to the **transformer backbone results reported in the original paper**.* The experiments in Table 1 indicate that our pre-training strategy enhances the performance of these state-of-the-art 3D detection methods. Furthermore, with the proposed SAM-guided tokenization and two-stage masked autoencoder, the performance improvement in VDETR is larger than in CAGroup3D, demonstrating the effectiveness of our method.
>
>
>
>
>
> | Methods    | Pre-trained | ScanNet ($AP_{25}$) | ScanNet ($AP_{50}$) |
> |------------|-------------|-----------------------|-----------------------|
> | CAGroup3D [4]  |  None     | 75.1                  | 61.3                  |
> | + Ours     | $\checkmark$            | **76.5**       | **62.4**       |
> | VDETR [5]     | None     | 73.6                  | 60.1                  |
> | + Ours     | $\checkmark$           | **75.8**       | **63.0**       |
>
> **Table 1:** 3D object detection results on ScanNet dataset based on CAGroup3D and VDETR.
>
>
>
> **Q2, comparison with Seal.**
>
> Please refer to the first part of the general response at the beginning.
>
>
>
> **Q3, comparison with other structure pre-training techniques like PPT.**
>
> Please refer to the second part of the general response at the beginning.
>
>
> **Q4, domain gap between the single-view point clouds and the complete room-level ones.**
>
> This is a great question. Previous 3D scene understanding pre-training strategies like PointContrast [6], DepthContrast [7], and PiMAE [8] all pre-train on single-view point clouds before fine-tuning on complete room-level point clouds. As demonstrated in PointContrast, *direct pre-training on complete room-level point clouds yields poorer results compared to single-view point clouds.* This may be due to several factors, including a more abundant and diverse set of training samples and the regularization effect of natural noise from camera instability.
>
> The limited domain gap between pre-training on single-view point clouds and fine-tuning on room-level multi-view point clouds can be attributed to several reasons. Pre-training on single-view point clouds helps the network learn detailed local features and geometric structures, which makes the network align well with the of room-level point clouds. This phase also enhances the network's robustness to various perspectives and partial visibility, facilitating effective knowledge transfer.
>
> **Q5, applicable to the online 3D perception setting.**
>
> Our method can also be applied to the online 3D perception setting. To demonstrate the generality of our approach, we applied it to the online 3D perception work [9]. However, due to differences in the backbone architecture, our pre-trained encoder cannot be directly integrated with the 3D detection method FCAF3D [10] used in [9]. Therefore, we replaced FCAF3D with VDETR [5] for the online detection. The experimental results in Table 2 below indicate that our method enhances performance in the online 3D perception setting, demonstrating the generality and effectiveness of the proposed pre-training approach.
>
>
>
> | Methods    | Pre-trained | ScanNet ($AP_{25}$) | ScanNet ($AP_{50}$) |
> |------------|-------------|-----------------------|-----------------------|
> | VDETR [5]  | *None*      | 73.6                  | 60.1                 |
> | VDETR-online  [9]   | *None*           | 68.9       | 52.7      |
> | VDETR-online + ours      | *$\checkmark$*       | **71.3**                  | **55.8**                |
>
> **Table 2:** Online 3D object detection results on ScanNet dataset.
>
> _________________
>
> [1] Segment Any Point Cloud Sequences by Distilling Vision Foundation Models, NeurIPS 2023
>
> [2] 2DPASS: 2D Priors Assisted Semantic Segmentation on LiDAR Point Clouds, ECCV 2022
>
> [3] LidarMultiNet: Towards a Unified Multi-Task Network for LiDAR Perception, AAAI 2023
>
> [4] CAGroup3D: Class-Aware Grouping for 3D Object Detection on Point Clouds, NeurIPS 2022
>
> [5] V-DETR: DETR with Vertex Relative Position Encoding for 3D Object Detection, ICLR 2024
>
> [6] PointContrast: Unsupervised Pre-training for 3D Point Cloud Understanding, ECCV 2020
>
> [7] Self-Supervised Pretraining of 3D Features on any Point-Cloud, ICCV 2021
>
> [8] PiMAE: Point Cloud and Image Interactive Masked Autoencoders for 3D Object Detection, CVPR 2023
>
> [9] Memory-based Adapters for Online 3D Scene Perception, CVPR 2024
>
> [10] FCAF3D: Fully Convolutional Anchor-Free 3D Object Detection, ECCV 2022

---

> > ### Comment · Reviewer_icGN · 2024-08-08
> >
> > The authors' rebuttal has solved most of my concerns. I suggest them to add more comparison and apply the proposed method to more architecture rather than only transformers. Also, the authors can include some visualization and demo on the online 3D perception setting, which shows great application potential.

---

> > > ### Author Response · Authors · 2024-08-09
> > >
> > > We sincerely appreciate your positive feedback. Your suggestions are invaluable in helping us further refine and enhance the quality of our paper.
> > >
> > > Regarding your suggestion to include more comparisons and apply the proposed method to a broader range of architectures beyond transformers, we conducted a related experiment in Table 1 of the rebuttal. Specifically, we applied our method to the CAGroup3D architecture, which has a different backbone design. However, it is important to note that a primary motivation behind our method is to address the misalignment between 2D and 3D representations caused by the traditional KNN tokenization method specifically within transformer structures. In future work, we plan to explore a more universal approach that better aligns 2D and 3D representations across all architectures.
> > >
> > > Additionally, we agree that the inclusion of more visualization results would enhance the demonstration of our method's potential in online perception settings. We will incorporate additional visualization results in the revised version to showcase these applications better.

---

### Official Review · Reviewer_DTv4 · 2024-07-12

**Soundness:** 3
**Presentation:** 2
**Contribution:** 3
**Rating:** 5
**Confidence:** 4

**Summary:**

This paper introduces a novel method to enhance 3D scene understanding by addressing misalignment between 2D and 3D representations and the long-tail distribution in 3D datasets. The proposed approach involves a SAM-guided tokenization method for seamless alignment and a group-balanced re-weighting strategy to handle representation imbalance. It also incorporates a two-stage masked token prediction process for effective knowledge distillation from 2D foundation models to 3D networks.

**Strengths:**

The SAM-guided tokenization method effectively aligns 2D and 3D representations, overcoming the limitations of traditional KNN-based tokenization techniques.

The introduction of a group-balanced re-weighting strategy addresses the long-tail distribution problem in 3D datasets, improving the representation of under-represented samples.

**Weaknesses:**

Mask generation is an important step of the proposed method, probably a visual comparison with current SOTA methods would be more helpful to justify its effectiveness.

There are typos in this manuscript, and the reference style is very confused (e.g., Bridge3D (10)).

**Questions:**

What is training time for fine-tuning? How about memory consumption compared to Bridge3D?

Bridge3D already mentioned in their paper that one of their limitations is that the current work primarily focuses on indoor 3D scene understanding. Is the same limitation applied for this paper?

**Limitations:**

Can this method apply to outdoor scene understanding?

---

> ### Author Rebuttal · Authors · 2024-08-06
>
> Thanks for your positive and insightful comments! We have addressed your questions as follows.
>
> **Q1, visual comparison with other mask generation methods.**
>
> We have added both visual and results comparisons with other mask generation strategies, such as superpoint and superpixel, to justify the effectiveness of SAM. Visualizations of these comparisons are included in the attached PDF file. Table 1 below indicates that utilizing SAM for mask generation achieves the best performance.
>
>
> | Masking strategy     | ScanNet ($AP_{25}$) | ScanNet ($AP_{50}$) |
> |---------------------|-----------------------|-----------------------|
> | Superpixel   | 66.0                 | 45.7                  |
> | Superpoint   | 66.8                  | 46.1              |
> | SAM | **68.2**                  | **48.4**                  |
>
> **Table 1:** Ablation study for different mask generation methods including superpixel, superpoint, and SAM.
>
>
>
>
>
> **Q2, what is training time for fine-tuning? How about memory consumption compared to Bridge3D?**
>
> We have included the training times in Table 2 below. Training Bridge3D requires approximately 31GB of memory, our method requires a total of 36GB of memory.
>
> | Methods       | SUN RGB-D | ScanNet |
> |---------------|-----------|-----------|
> | 3DETR         | 29 hours  | 11 hours  |
> | GroupFree3D   | 15 hours  | 4 hours   |
>
> **Table 2:** Fine-tuning times for 3DETR and GroupFree3D on SUN RGB-D and ScanNetV2 datasets with 2 A100 GPUs.
>
>
> **Q3, application in the outdoor scene understanding.**
>
> One key motivation for our method is to address the 2D-3D alignment challenges posed by traditional KNN-based tokenization methods. In outdoor scenarios, point clouds are often very sparse, especially at long-range distances, making KNN-based tokenization methods unsuitable. Current transformer-based methods for outdoor environments utilize voxel or pillar methods for feature extraction, treating these voxels or pillars as 2D images. Due to significant differences between outdoor and indoor environments and the sparsity of point clouds, our SAM-guided tokenization may not be suitable for outdoor scenarios. However, other parts of our method, such as representation re-weighting and two-stage masked token prediction, can be effectively applied in outdoor contexts.
>
> We have chosen BEV-MAE [2] as our baseline method to distill 2D features into 3D BEV maps using the proposed re-weighting strategy. In the second stage, we enable the student network to reconstruct the masked BEV features obtained from the first-stage teacher models. As shown in Table 3 below, our re-weighting strategy and two-stage masked token prediction also benefit outdoor scenarios. Due to time constraints, we were unable to fully optimize the training parameters, indicating that there is still room for performance improvement.
>
>
>
> | Methods    | Waymo ($mAP$) | Waymo ($APH$) |
> |------------|---------------|---------------|
> | Scratch    | 65.60         | 63.21         |
> | GD-MAE [1] | 66.9          | 64.53         |
> | BEV-MAE [2]| 67.02         | 64.55         |
> | Ours       | **67.81**     | **64.93**     |
>
> **Table 3:** 3D detection results in the outdoor Waymo dataset.
>
> **W1, typos, and format issues.**
>
> Thanks for pointing out! We will correct the typos and update the reference style in the revised version of the paper.
>
> _________________
>
>
> [1] GD-MAE: Generative Decoder for MAE Pre-training on LiDAR Point Clouds, CVPR 2023
>
> [2] BEV-MAE: Bird's Eye View Masked Autoencoders for Point Cloud Pre-training in Autonomous Driving Scenarios, AAAI 2024

---

> > ### Comment · Reviewer_DTv4 · 2024-08-11
> >
> > Thanks for the rebuttal, most of my concerns are properly addressed.

---

> > > ### Author Response · Authors · 2024-08-11
> > >
> > > We sincerely appreciate your positive feedback. Your suggestions are invaluable in helping us further refine and enhance the quality of our paper.

---

### Official Review · Reviewer_EQZo · 2024-07-13

**Soundness:** 3
**Presentation:** 2
**Contribution:** 3
**Rating:** 7
**Confidence:** 3

**Summary:**

The paper proposes a 3D transformer tokenization technique to align 3D representations with 2D ones, distilling from 2D pre-trained knowledge from SAM. The method achieves favorable performance on 3D object detection and semantic segmentation compared to prior self-supervised learning methods.

**Strengths:**

* The paper provides a good explanation contrasting the proposed method with prior works that develop 3D representations distilling from 2D foundation models, highlighting the drawbacks of KNN-based point tokenization which well motivates the proposed solution.
* Several techniques, such as group-balanced re-weighting and two-stage teacher-forcing training, are adopted in the framework and ablations show their effectiveness.
* The performance is strong, with advantages compared to prior self-supervised methods.

**Weaknesses:**

* It is not discussed why SAM is chosen as the distillation source for tokenization, instead of other 2D foundation models such as DINO, or even generative ones such as Stable Diffusion.
* The representation is evaluated on the task of 3D object detection and 3D semantic segmentation, which intuitively seem to correspond well to the strength of SAM as SAM may have implicitly learned the notion of objects and object semantics during its training phase. Are there tasks where the proposed representation could be useful?

**Questions:**

* What's the input to $F_{2D, i}$ and $F_{3D, i}$ in Eq (1)? Where is $i$ sampled from?
* $1/O_i$ in Eq (1) should probably be written as $1/|O_i|$.
* What is $n_{min}$ and $n_{max}$ in line 209?

**Limitations:**

Limitations are discussed.

---

> ### Author Rebuttal · Authors · 2024-08-06
>
> Thanks for your positive and insightful comments! We have addressed your questions as follows.
>
>
> **Q1, why SAM is chosen as the distillation source for tokenization**:
>
> We chose SAM [1] to guide tokenization because it generates great zero-shot masks that provide boundary regularities and effectively facilitate region-level knowledge distillation. In contrast, other foundational models, such as Stable Diffusion [2] or DINOv2 [3], cannot generate comparable zero-shot masks.
>
> For feature distillation, we use DINOv2 as the teacher model, enabling the student 3D models to predict the same region-level features obtained from DINOv2. We conducted an ablation study comparing features obtained from other foundational models like SAM, Stable Diffusion, and CLIP [4] to extract visual features. The results in Table 1 below indicate that utilizing DINOv2 achieves the best performance.
>
>
> | Foundation Models | ScanNet ($AP_{25}$)  | ScanNet($AP_{50}$)  |
> |-------------------|---------------------|---------------------|
> | Stable Diffusion [2] | 66.2                | 45.7               |
> | CLIP [4]          | 66.8               | 46.3                |
> | SAM [1]           | 67.5                | 46.7                |
> | DINOv2 [3]        | **68.2**            | **48.4**            |
>
> **Table 1:** Ablation study for the choice of foundation models for representation distillation.
>
>
>
>
>
> **Q2: other tasks where the proposed representation could be useful.**
>
>
> Reviewer icGN suggests that our method could also be applied in online 3D perception settings. To demonstrate the generality of our approach, we applied it to the online 3D perception method described in [5]. However, due to differences in the backbone architecture, our pre-trained encoder cannot be directly integrated with the 3D detection method FCAF3D [6] used in [5]. Therefore, we replaced FCAF3D with the transformer-based VDETR [7] for online detection. The experimental results in Table 2 indicate that our method enhances performance in the online 3D perception setting, demonstrating the generality and effectiveness of the proposed pre-training approach.
>
> Additionally, by replacing the DINOv2 teacher model with CLIP, our method could potentially enable 3D zero-shot semantic segmentation by calculating the feature similarity between the predicted features and the CLIP text features. Furthermore, since our method distills features from 2D foundation models into 3D networks, it can enhance various 3D tasks that benefit from 2D features, such as 3D scene classification, 3D reconstruction, and 3D instance segmentation. However, due to time limitations, we were unable to apply our method to these tasks in the current study, but we plan to do so in the future to further demonstrate the generality of the proposed method.
>
> | Methods    | Pre-trained | ScanNet ($AP_{25}$) | ScanNet ($AP_{50}$) |
> |------------|-------------|-----------------------|-----------------------|
> | VDETR-online  [5]   | *None*           | 68.9       | 52.7      |
> | VDETR-online + ours      | *$\checkmark$*       | **71.3**                  | **55.8**                |
>
> **Table 2:** Online 3D object detection results on ScanNet dataset.
>
>
> **Q3, some definitions.**
>
> **What is the input to the $F_{2D,i}$ and $F_{3D,i}$?**
>
> As illustrated in the main paper (lines 174 to 177), $I$ represents pixel-level features obtained from raw images processed by foundation models and feature interpolation. $H_i$ represents the SAM-guided tokenized features obtained from the raw point clouds with the $i$-th masks generated by SAM. In Equation 1, we utilize the $i$-th masks to group the pixel-level image feature representations $I$ into the $i$-th object-level features to obtain $F_{2D,i}$ and utilize one MLP project layer to project $H_i$ to the $F_{3D,i}$. Therefore, the inputs for $F_{2D,i}$ and $F_{3D,i}$ are the raw images, point clouds, and the $i$-th masks generated by SAM.
>
> **Where is the i sampled from?**
>
> $i$ represents the $i$-th masks generated by SAM.
>
> **What is $n_{min}$ and $n_{max}$?**
>
> $n_{min}$ and $n_{max}$ represent the minimum and maximum numbers of feature clusters.
>
> **$\frac{1}{O_i}$ in Eq (1) should probably be written as $ \frac{1}{|O_i|}$**
>
> Thanks for pointing out! We will correct $\frac{1}{O_i}$ in equation (1) to $ \frac{1}{|O_i|}$.
>
> _________________
>
> [1] Segment Anything, CVPR 2023
>
> [2] High-Resolution Image Synthesis with Latent Diffusion Models, CVPR 2022
>
> [3] DINOv2: Learning Robust Visual Features without Supervision, TMLR 2024
>
> [4] Learning Transferable Visual Models From Natural Language Supervision, ICML, 2021
>
> [5] Memory-based Adapters for Online 3D Scene Perception, CVPR 2024
>
> [6] FCAF3D: Fully Convolutional Anchor-Free 3D Object Detection, ECCV 2022
>
> [7] V-DETR: DETR with Vertex Relative Position Encoding for 3D Object Detection, ICLR 2024

---

> ### Comment · Area_Chair_Bw8K · 2024-08-12
> **Discussion**
>
> Thank you for being a reviewer for NeurIPS2024, your service is invaluable to the community!
>
> The authors have submitted their feedback.
>
> Could you check the rebuttal and other reviewers' comments and start a discussion with the authors and other reviewers?
>
> Regards,
> Your AC

---

### Official Review · Reviewer_KyqU · 2024-07-13

**Soundness:** 2
**Presentation:** 3
**Contribution:** 2
**Rating:** 5
**Confidence:** 5

**Summary:**

The paper proposes a self-supervised method for understanding 3D scenes by predicting the 3D mask of the point cloud. The masks are initialized from Segment Anything (SAM), followed by two stages of the knowledge distillation framework to train the 3D teacher and student networks. The method is evaluated on SUN RGB-D, ScanNet, and S3DIS datasets.

**Strengths:**

The paper is generally well-written and easy to follow.

**Weaknesses:**

- The first concern is the method's innovation. Other methods, like Seal [P1], have already proposed the idea of distilling the knowledge from 2D foundation models into the 3D network for mask prediction. However, it is not discussed or compared.

- The insight into method design is not elaborated. It is not clear why two-stage training is necessary for self-supervised learning. How about three-stage, four-stage, or momentum updating proposed in MeanTeacher? It is better to present more insight and intuitive explanations. Besides, it is difficult to understand why group-balanced re-weighting will work since the pseudo labels are also extremely noisy, i.e., it also suffers from long-tail issues.
- More recent methods should be comparied to verify the efficientness of the proposed method. For example, CLIP2Scene is disscussed but not comparied. Seal [P1], also utilize 2D models mask prediction to regularize 3D network.

P1. Segment Any Point Cloud Sequences by Distilling Vision Foundation Models, NeurIPS 2023.

**Questions:**

refer to the weaknesses

**Limitations:**

yes

---

> ### Author Rebuttal · Authors · 2024-08-06
>
> We sincerely appreciate your thorough review and valuable suggestions. We have addressed your questions as follows.
>
>
>
> **Q1, innovation of method.**
>
> Please refer to the first part of the general response at the beginning.
>
>
> **Q2, the insight of the two-stage design.**
>
>
> Masked token prediction has proven effective for single-modality learning [4]. Our method extends this concept to cross-modality knowledge distillation by designing an efficient two-stage framework for cross-modality masked feature prediction. Our two-stage design works as follows: *In the first stage*, we perform knowledge distillation using SAM-guided tokenization and SAM masks to seamlessly transfer 2D region-level information to 3D. *In the second stage*, we use masked-view inputs to predict contextualized 3D representations within a latent space aligned by a teacher model with complete-view inputs. *This ensures that student models learn well-aligned and contextualized representations*, leveraging the proven efficiency of masked feature prediction for multi-modality self-supervised learning.
>
> The ablation study presented in Table 5 of the main paper (also shown in Table 1 below) demonstrates that the two-stage design outperforms the one-stage. In the one-stage design, the 3D encoder directly predicts masked parts of the 2D features from foundation models, resulting in poorer performance. This underperformance is likely due to the significant domain gap between 2D and 3D models, which causes the network to learn sub-optimal features when applying one-stage masked 2D feature prediction.
>
> Our two-stage design addresses this issue by separating the process into two distinct phases: the first stage focuses on 2D to 3D knowledge distillation, and the second stage handles 3D masked token prediction. This approach effectively reduces the domain gap between 2D and 3D representations, leading to better representations and improved performance.
>
> **Q3, reasons for two stage design instead, three, or even four stages.**
>
> In the second stage, the teacher model is frozen, and the student model is trained to predict the masked feature tokens. Since the teacher model's weights remain unchanged, adding more stages would only extend the training duration for the student encoder in the second stage without providing additional benefits. Therefore, adding additional stages to train the student encoder, such as a third or fourth, is unnecessary. The experiments shown in Table 1 below indicate that introducing a third or fourth stage yields similar results to the two-stage setting even with more training time.
>
> | Stage settings      | ScanNet ($AP_{25}$) | ScanNet ($AP_{50}$) |
> |---------------------|-----------------------|-----------------------|
> | One-stage setting   | 66.0                  | 46.3                  |
> | Two-stage setting   | 68.2                  | **48.4**              |
> | Three-stage setting | 67.7                  | 48.3                  |
> | Four-stage setting  | **68.4**              | 47.5                  |
>
> **Table 1:** Ablation study for stage settings.
>
>
>
>
>
> **Q4, why not utilize momentum updating?**
>
> When utilizing momentum updating, two augmented positive views are sent to both the trainable encoder and the momentum encoder to obtain positive representation pairs or pseudo-label pairs. The weights of the momentum encoder are then updated with the weight of the trainable encoder through the contrastive loss or the supervision loss via pseudo-labels. However, our method does not require two augmented views for training in either stage; therefore, momentum updating is not applicable to our approach.
>
>
> **Q5, why group-balanced re-weighting will work.**
>
> Group-balanced re-weighting is introduced to address the long-tail problem inherent in the natural imbalance of object class occurrences. As discussed in the paper [5], *foundation models provide well-represented patch-level features that can effectively identify different classes through clustering-based methods due to their training on large datasets, making it effective as a zero-shot learner.*
>
>
> In this paper, we take advantage of the foundational model to extract object-level features and apply a clustering method to compute distribution statistics. These statistics are less noise-prone because the foundational model provides robust feature representation for both head and tail class objects. We then normalize the distribution factors as weights to regularize the distillation process, offering an effective solution to the long-tail issue.
>
>
> **Q6, comparison with CLIP2Scene and Seal.**
>
> Please refer to the second part of the general response at the beginning.
> _________________
>
>
>
> [1] Segment Any Point Cloud Sequences by Distilling Vision Foundation Models, NeurIPS 2023
>
> [2] Bridging the domain gap: Self-supervised 3d scene understanding with foundation models, NeurIPS 2023
>
> [3] Videoprism: A foundational visual encoder for video understanding, ICML 2024
>
> [4] Self-supervised learning from images with a joint-embedding predictive architecture, CVPR 2023
>
> [5] Deep ViT Features as Dense Visual Descriptors, ECCVW 2022

---

> > ### Comment · Reviewer_KyqU · 2024-08-12
> > **reply to authors**
> >
> > Thanks for the rebuttal, most of my concerns are addressed.

---

> > > ### Author Response · Authors · 2024-08-12
> > >
> > > We sincerely appreciate your positive feedback. Your suggestions are invaluable in helping us further refine and enhance the quality of our paper.

---

### Author Rebuttal · Authors · 2024-08-06

**General Response**
--
We sincerely thank each reviewer for their thoughtful feedback and detailed reviews. We address the main concerns regarding novelty and comparisons with other peer research below.

**Comparison with Seal and the innovation of methodology.**

Thank you for highlighting Seal [1] as a related work. We did not discuss and compare it with our work because *Seal uses a different backbone (3D U-Net), is pre-trained on different datasets (outdoor scenes), and is fine-tuned only for 3D semantic segmentation tasks.* 3D U-Net struggles with scalability and flexibility, making it less effective for scaling and handling tasks such as detection. Furthermore, we have already conducted a detailed comparison with the most relevant work, Bridge3D [2], which employs a similar strategy to Seal’s by leveraging SAM masks and 2D features. Both Bridge3D and Seal directly use SAM masks to group 2D and 3D object-level features for object-level contrastive learning. We discuss the differences between our method and these SAM-guided group contrastive learning methods in lines 32 to 47 of the main paper. **Notably, our dense distillation setting in the ablation study shown in Table 3 of the main paper is the same as the Seal method**, except that Seal utilizes the InfoNCE loss, whereas dense distillation leverages the SmoothL1 loss. To further demonstrate the effectiveness of our method, we will cite and compare Seal in the revised version of our paper. The comparison results with Seal are included in Table 1 below.

Compared to Seal, our method is unique in *addressing two critical challenges*: the misalignment of object-level 2D and 3D features for  traditional KNN-based tokenization methods, and the imbalance problem of representations during the distillation. *To solve the first issue*, we propose a SAM-guided 3D tokenization method that seamlessly aligns 2D and 3D object-level features. *For the second challenge*, we introduce a self-supervised learning re-weighting strategy to enhance the weight of tail representations. *Furthermore, another novel aspect* of our approach is extending previous single modality masked token prediction into cross-modality learning with a two-stage framework design to learn well-aligned and contextualized 3D representations.


**Results comparison with pre-training methods for other backbones like Seal [1], PPT [3], and CLIP2Scene [4].**

In the main paper, we did not compare our results with Seal [1], PPT [3], and CLIP2Scene [4] as they utilize 3D-UNet as the backbone and are fine-tuned exclusively for 3D semantic segmentation tasks. Most previous object-level and scene-level 3D transformer-based pre-training methods, such as I2P-MAE [5], PiMAE [6], and Bridge3D [2], *do not directly compare their approaches with other pre-training strategies based on other 3D backbones like PointNet, DGCNN, or 3D-UNet.* Hence, we followed the same comparison strategy.

However, we recognize that including comparisons with methods using other backbones could better illustrate the effectiveness of our approach. Therefore, we applied the methodologies of Seal, PPT, and CLIP2Scene to the transformer structure and utilized the same settings as our method. As shown in Table 1 below, our method achieves the best performance, highlighting the advantages of our proposed strategies. In the revised version, we will cite those papers and include a discussion of their methodologies and results.


| Methods        | ScanNet ($AP_{25}$) | ScanNet ($AP_{50}$) | ScanNet ($mIoU$) |
|----------------|-----------------------|-----------------------|--------------------|
| Scratch        | 61.1                  | 38.6                  | 67.3               |
| CLIP2Scene [4] | 62.0                  | 40.1                  | 69.2               |
| Seal [1]       | 62.7                  | 41.3                  | 70.3               |
| PPT [3]        | 62.8                  | 42.1                  | 70.9               |
| Bridge3D [2]   | 65.3                  | 44.2                  | 73.9               |
| Ours           | **68.2**              | **48.4**              | **75.4**           |

**Table 1:** Comparison with other state-of-the-art pre-training methods on ScanNet dataset with 3D detection and semantic segmentation tasks.


The attached PDF contains a table and a figure, where the figure presents a visual comparison with other mask generation methods to address the question from Reviewer DTv4. Additionally, the table includes the ablation study results of different mask-generation strategies.

_________________


[1] Segment Any Point Cloud Sequences by Distilling Vision Foundation Models, NeurIPS 2023

[2] Bridging the domain gap: Self-supervised 3d scene understanding with foundation models, NeurIPS 2023

[3] Towards Large-scale 3D Representation Learning with Multi-dataset Point Prompt Training, CVPR 2024

[4] Towards Label-efficient 3D Scene Understanding by CLIP, CVPR 2023

[5] Learning 3D Representations from 2D Pre-trained Models via Image-to-Point Masked Autoencoders, CVPR 2023

[6]PiMAE: Point Cloud and Image Interactive Masked Autoencoders for 3D Object Detection, CVPR 2023

---

### Decision · Program_Chairs · 2024-09-25

**Decision:**

Accept (poster)

**Comment:**

The paper received three Borderline accepts and one Accept after the rebuttal and discussion period. All reviewers acknowledged the novelty and effectiveness of the proposed method, and the AC found no reasonable cause to overturn all of the reviewers' positive evaluations. Therefore, considering the submitted paper, the reviewers' comments, the authors' rebuttals, and the discussion, the AC finds it reasonable to conclude that the paper is acceptable. However, the AC recommends a clearer description of the novelty of the method and qualitative and quantitative comparisons with state-of-the-art methods such as Seal.